# The Role of Increased Expression of Sirtuin 6 in the Prevention of Premature Aging Pathomechanisms

**DOI:** 10.3390/ijms24119655

**Published:** 2023-06-02

**Authors:** Adrianna Dzidek, Olga Czerwińska-Ledwig, Małgorzata Żychowska, Wanda Pilch, Anna Piotrowska

**Affiliations:** 1Doctoral School of Physical Culture Science, University of Physical Education, 31-571 Krakow, Poland; 2Institute for Basic Sciences, Faculty of Physiotherapy, University of Physical Education, 31-571 Krakow, Poland; 3Faculty of Health Sciences and Physical Culture, Biological Fundation of Physical Culture, Kazimierz Wielki University in Bydgoszcz, 85-064 Bydgoszcz, Poland

**Keywords:** sirtuins, SIRT6, gene expression, aging, inflammation, metabolism, aging pathomechanism, physical activity

## Abstract

Sirtuins, in mammals, are a group of seven enzymes (SIRT1–SIRT7) involved in the post-translational modification of proteins—they are considered longevity proteins. SIRT6, classified as class IV, is located on the cell nucleus; however, its action is also connected with other regions, e.g., mitochondria and cytoplasm. It affects many molecular pathways involved in aging: telomere maintenance, DNA repair, inflammatory processes or glycolysis. A literature search for keywords or phrases was carried out in PubMed and further searches were carried out on the ClinicalTrials.gov website. The role of SIRT6 in both premature and chronological aging has been pointed out. SIRT6 is involved in the regulation of homeostasis—an increase in the protein’s activity has been noted in calorie-restriction diets and with significant weight loss, among others. Expression of this protein is also elevated in people who regularly exercise. SIRT6 has been shown to have different effects on inflammation, depending on the cells involved. The protein is considered a factor in phenotypic attachment and the migratory responses of macrophages, thus accelerating the process of wound healing. Furthermore, exogenous substances will affect the expression level of *SIRT6*: resveratrol, sirtinol, flavonoids, cyanidin, quercetin and others. This study discusses the importance of the role of SIRT6 in aging, metabolic activity, inflammation, the wound healing process and physical activity.

## 1. Introduction

Silent information regulator (Sir) proteins belong to NAD^+^-dependent deacetylases, enzymes catalyzing the deacetylation reaction. Sirtuins were first discovered in the yeast *Saccharomyces cerevisiae* by virtue of their role in the establishment of transcriptional silencing of mating-type loci. Further studies have shown that Sir2 is also crucial in the partitioning of carbonylated proteins between mother and daughter cells, as well as for silencing at yeast telomeres and in the rDNA. The conserved enzymes called sirtuins, originally discovered in yeast, are produced by almost all organisms, from non-nucleated prokaryotes, to unicellular archaea and bacteria, to mammals [1,2].

Sirtuins play an important role in maintaining health and affect many pathways that increase the lifespan of organisms [1]. In mammals, they constitute a group of seven proteins (SIRT 1–7) belonging to class III histone deacetylases (HDACs). Sirtuins have a common catalytic domain of NAD^+^, consisting of about 260 amino acid residues [2]. Individual sirtuin isoforms differ in sequence and length in the N- and C-terminal domains, which affects their enzymatic activity, cellular localization and substrate specificity [3].

Initially, sirtuins were studied in the context of organism aging and gene silencing, but many other biological functions of proteins belonging to this group have been revealed in mammalian cells [4]. Studies on yeast aging showed a relationship between the *Silent Information Regulator 2 (Sir2)* gene and the viability of budding yeast [5].

Sirtuin 6 (SIRT6), belonging to class IV, is in the cell nucleus. SIRT6 promotes the repair of double-stranded DNA breaks by forming a complex with DNA-dependent protein kinase (DNA-PK) [6]. It has a hydrophobic pocket where the myristoylated group is located before being cleaved from the modified protein [7,8]. It has been suggested that this enzyme exhibits higher deacylase activity than deacetylase activity [4].

In a healthy organism or with minor damage, this protein promotes cell proliferation and activation of repair processes, while in the case of serious damage, it supports the apoptosis of damaged cells. It is in this mechanism that SIRT6 prevents the proliferation of damaged cancer cells [9,10]. SIRT6 slows down the course of gluconeogenesis by inhibiting the action of the PGC-1α factor. The activity of PGC-1α depends on the degree of its acetylation. This process is controlled by the acetyltransferase GCN5. SIRT6 causes deacetylation and phosphorylation of GCN5, which increases its acetylase activity. Over-acetylating PGC-1α, the acetyltransferase leads to a decrease in its activity and, consequently, to the inhibition of gluconeogenesis [11,12].

The association of sirtuins with aging is a well-known topic. Imai et al. [13] pointed to several mechanisms, including: effects on metabolism and regulation of circadian clock mechanisms, and emphasized that sirtuin activity decreases with age, which should be associated with a decrease in NAD^+^. The most important sirtuin described by the paper’s authors was SIRT1. Furthermore, it has been the most extensively studied protein of the mammalian sirtuin family. However, it has been shown that not only a loss of SIRT1 activity enhances DNA damage, but also SIRT6 [14]. Therefore, the authors of this review considered it interesting to examine the significance of SIRT6 in the aging process. The mechanisms of SIRT6 action indicate that this protein should be considered as an important element coordinating processes related to maintaining homeostasis and thereby inhibiting the premature aging of the organism. In this respect, knowledge of the mechanisms of SIRT6 action and factors modifying its expression becomes an interesting issue for a number of specialists. Therefore, the aim of this study was to indicate how this protein acts in a healthy organism, what changes are observed in pathological processes and what internal and external stimuli modify the process of *SIRT6* expression.

## 2. Structure

SIRT6 is a widely expressed protein [15]. Mahlknecht et al. [15] demonstrated that the human *SIRT6* genomic sequence has one single genomic locus, which spans a region of 8427 bp. The human *SIRT6* gene is located on chromosome 19p13.3 and consists of eight exons ranging in size from 60 bp (exon 4) to 838 bp (exon 8). The mRNA for human SIRT6 encodes a protein of 355 amino acids, with a predicted molecular weight of 39.1 kDa and an isoelectric point of 9.12 [9]. SIRT6 is composed of two main domains—a large domain containing a nucleotide binding and Rossmann fold (responsible for binding NAD^+^) and a small domain containing a Zn^2+^ binding loop. Zinc ions play a role in maintaining the integrity of the catalytic domain and stabilizing the enzyme structure [16].

### Activation and Reactivation of Gene Expression

The DNA molecule, which is a long strand, is wound around histone proteins that constitute a structural element of chromatin. The modification reaction of histone proteins is histone deacetylation, which involves the detachment of an acetyl group from the N-terminus of the histone. This reaction is carried out by histone deacetylases [17].

SIRT6, as a histone deacetylase, inhibits the transcriptional activity of several transcription factors and deacetylates specific histone lysine residues, such as H3K9 and H3K56, depending on NAD^+^ [18].

The N-terminal extension of SIRT6 is important for chromatin association and the internal catalytic activity of the core domain [16]. As a non-histone protein deacetylase, SIRT6 deacetylates forkhead box O1 protein, pyruvate kinase M2, C-terminal binding protein and GATA-binding protein 3 [18]. The C-terminal extension is essential for proper nuclear localization [16].

Although SIRT6-mediated histone deacetylation generally correlates with chromatin condensation and gene silencing [19,20], there is also evidence that SIRT6 can activate certain genes by mediating histone deacetylation. For example, SIRT6 acts as a transcriptional coactivator of erythroid 2-related factor 2 (Nrf2) to protect against oxidative stress in human mesenchymal stem cells, and it has been found that SIRT6 is in a protein complex with Nrf2 and RNAP II [21,22]. Nrf2 is one of the master regulators of antioxidant responses—binds to the antioxidant response elements (AREs) and activates antioxidant genes, for example, heme oxygenase 1 (HMOX1), which is known for counteracting reactive oxygen species (ROS). In aging cells, a decrease in Nrf2-ARE activity is observed, which may be caused by oxidative stress-related tissue degeneration [21]. Nrf2 activation occurs during evasion from Keap1-mediated proteasomal degradation in the cytosol [23]. SIRT6 activates Nrf2 by inhibiting Keap1 transcription and directly interacting with Nrf2 [24]. It is indicated that SIRT6 is required for HOMX1 induction in response to oxidative stress and SIRT6 deficiency increases basal cellular ROS [21] (Figure 1).

## 3. Aging

Aging of the body is a physiological, dynamic and inevitable process. It is believed that the aging process begins at different times in different systems (the skin is one of the first organs to age). However, aging usually begins in the fourth decade of life and ends at the end of biological life. Important factors influencing aging are extrinsic, biological and psychosocial factors. However, the basic determinant of organism aging is the individual’s genotype [25]. Lopez-Otin et al. [26] proposed nine so-called hallmarks of aging: telomere attrition, genomic instability, loss of proteostasis, epigenetic alterations, deregulated nutrient sensing, mitochondrial dysfunction, stem cell exhaustion, cellular senescence and altered intercellular communication. There are some symptoms that occur in psychological aging, but at a younger age [27]. It is associated with accelerated or premature aging, where interrelated molecular and cellular phenomena are intensified.

Nowadays, it is hypothesized that SIRT6 regulates lifespan by influencing a series of processes that control aging, such as genome stability, transcriptional processes of DNA repair elements, the ability to regulate telomere length and metabolic homeostasis (including carbohydrate metabolism) [9,28,29] (Figure 2).

Mostoslavsky et al. [30] demonstrated that mice deficient in SIRT6 are small and develop dysfunctions related to the loss of subcutaneous fat, lordokyphosis and severe metabolic defects by 2–3 weeks of age. These symptoms can be considered similar to those associated with premature aging. These mice eventually died after about 4 weeks. On the other hand, Kanfi et al. [8] observed an extended lifespan of transgenic mice with overexpression of SIRT6 compared to wild-type mice. Interestingly, a significant difference was only observed in males. The authors suggested that this may be related to lower levels of insulin-like growth factor 1 (IGF1) in serum, altered levels of phosphorylation of major IGF1 signaling components and higher levels of IGF1 binding protein, which affects lifespan regulation [31].

Betry et al. [32] conducted a study on 14 men and 12 women in their sixth and seventh decades of life with similar anthropometric and metabolic parameters. The participants underwent six tests: mini-mental state examination—a test that allows for the detection of cognitive dysfunction; Doubis’ five-word test—a screening memory task specifically built to discriminate hippocampal memory deficits; Chair Stand Test—allowing for the detection of fall risk in senior patients; Time Up&Go, walking speed and balance tests—allowing for the quantifiable assessment of motor function and serving as an important observation of functional capacity, Center for Epidemiologic Studies Depression scale—CES-D; and Cognitive Difficulties Scale—CDS. Based on the results obtained in the above tests, patients were assigned to the frail group (first tercile of the index score) or the robust group (individuals from the third tercile). The researchers [32] indicated that skeletal muscle biopsies taken from individuals in the robust group showed significantly lower expression of *SIRT6*. However, no differences were observed in the level of expression of this protein in leukocytes isolated from the patients’ blood. In the Betry et al. [32] study, an interesting correlation was also noted. The levels of *CPT1b* mRNA are strongly correlated with SIRT6 levels in skeletal muscle. The authors suggested that the overexpression of *CPT1b* in the robust group may at least partially explain their better metabolic profile, and the significant inverse correlation with the level of *SIRT6* expression in muscle could be associated.

SIRT6 is a central regulator of mitochondrial activity in the brain. Deficiency of this protein in the central nervous system leads to mitochondrial deficiency with marked changes in metabolite content and global downregulation of mitochondria-related genes [33]. The metabolomic changes observed in SIRT6 deficiency lead to an increase in ROS generation. Smirnov et al. [33] indicated that the deficit of SIRT6 in the brain occurs during the aging process of the human brain, especially in patients with Parkinson’s, Alzheimer’s, Huntington’s and Amyotrophic lateral sclerosis disease.

Telomeres shorten with increased turnover and chronological age in somatic stem cells. In early adult life, telomeres help to stabilize the nuclear genome, whereas, in post-reproductive age, they decrease the intensity of age-related changes [34]. Michishita et al. [35] indicated that SIRT6 is associated with specific telomeres, and cells with reduced SIRT6 activity show telomere structures resembling defects observed in Werner syndrome. The Werner syndrome protein (WRN) plays an important role in telomere metabolism and during DNA replication. The connection between WRN and chromatin is closely related to SIRT6-dependent deacetylation of telomeric residues H3K9 and H3K59. Genomic instability with loss of SIRT6 may thus be related to loss of WRN-chromatin connection and influence the redirection of the cell towards premature aging [36]. The cited researchers [35] concluded that SIRT6 contributes to maintaining the specialized chromatin state in mammalian telomeres, which in turn is required for proper telomere metabolism and function. Michishita et al.’s study links SIRT6’s chromatin regulation with telomere maintenance and premature aging syndrome [35].

Another mechanism in which *SIRT6* expression will interfere with the aging process is its influence on retrotransposon activity. Transposable elements are DNA sequences capable of integration into other genomic locations [12]. Class I retrotransposons transport via intermediate RNA. Balestrieri et al. [37] identified L1 retrotransposons as a class of transcriptional elements that strongly mediate genomic instability and play a role in age-related pathologies. SIRT6 is a strong repressor of L1 activity, binding to the 5’ UTR of L1 loci. In response to DNA damage and aging, SIRT6 has a reduced ability to bind to L1 loci, contributing to the activation of previously silenced retroelements [38].

Skin aging is the first and most visible sign of aging in the body. Technical advancements in recent years have allowed for a broader examination of the aging process in various types of skin cells. Fibroblasts, as the most numerous types of cells in the dermis, are directly or indirectly responsible for most skin aging characteristics [39]. Sharma et al. [6] demonstrated significantly decreased *SIRT6* expression in fibroblasts of older individuals (above 50 years old) compared to younger individuals (below 18 years old). Furthermore, it was shown that fibroblasts of older individuals were more resistant to reprogramming into induced pluripotent stem cells (iPSCs) compared to fibroblasts of younger individuals. The inclusion of SIRT6 with classical Yamanaka factors in older fibroblasts improved their reprogramming efficiency, which was then similar to that of younger individuals [6].

MicroRNAs (miRNAs) are a class of non-coding RNAs that play a significant role in gene regulation; modulating numerous biological processes; including aging. The influence of miRNAs on the post-transcriptional regulation of SIRT6 is still being studied. Sharma et al. [6] confirmed that SIRT6 regulates the transcription of miR-766, and blocking miR-766 significantly improves the efficiency of reprogramming aging cells. Higher levels of miR-766 in older fibroblasts were inversely correlated with SIRT6 levels [6]. Increasing SIRT6 activity in fibroblasts may therefore slow down skin aging.

An important factor accelerating the aging of the organism is an unhealthy diet and low levels of physical activity and their far-reaching consequences in the form of metabolic syndrome. SIRT6 is indicated as a potential therapeutic target against metabolic syndrome. This protein is associated with calorie reduction. Activating SIRT6 is also suggested to have potentially beneficial effects on age-related metabolic diseases. The development of small molecule *SIRT6* activators could therefore have great therapeutic potential [40].

## 4. Metabolic Activity

SIRT6 is involved in regulating glucose homeostasis in the body. It has been shown that SIRT6 levels are increased during fasting. Additionally, by increasing the expression of genes involved in gluconeogenesis, it controls this process in the liver. Zhong et al. [41] discovered that the expression of gluconeogenic genes was increased in livers with SIRT6 deficiency. Researchers also identified the role of SIRT6 as a corepressor of Hif1α (Hypoxia-inducible factor 1α), a critical regulator of the response to nutritional stress. Cells deficient in SIRT6 exhibit increased Hif1α activity, thus showing increased glucose uptake. SIRT6, therefore, acts as a corepressor of the transcription factor Hif1α, reducing glycolysis during normal nutrition and stimulating mitochondrial fatty acid oxidation [41].

Kim et al. [42] pointed out that liver-specific deletion of *SIRT6* in mice leads to profound changes in gene expression, causing increased glycolysis, intensified triglyceride synthesis, decreased β-oxidation and liver steatosis. Other authors suggest that SIRT6 may therefore be a potential target in the treatment of liver diseases characterized by lipid accumulation [43].

Dominy et al. [11] indicated the usefulness of activating liver *SIRT6* in the therapeutic treatment of insulin-resistant diabetes. Xiong et al. [44] demonstrated a significant decrease (50%) in glucose-stimulated insulin secretion in mice subjected to *SIRT6* knockdown in pancreatic beta cells. The mice also had lower levels of ATP in the studied cells compared to the wild-type control group. An increased number of damaged mitochondria was also identified. Based on the obtained results, the authors suggested that SIRT6 regulates proper insulin secretion through the regulation of mitochondrial glucose oxidation. Therefore, activating the protein may be helpful in improving insulin secretion in diabetic states. The process described here takes place in the mitochondria. This underscores that SIRT6 acts in many cellular organelles, so it can be said that it acts throughout the cell.

Further experiments have indicated that mice with *SIRT6* knockout were more susceptible to high-fat diet-induced obesity, attributed to adipocyte hypertrophy. Moreover, increased macrophage infiltration into the examined adipose tissue was observed, indicating an intensified inflammatory process in these mice. It was found that SIRT6 regulates energy homeostasis by modulating the activity and expression of lipase in adipose tissue [45]. Kanfi et al. [40] noticed increased glucose tolerance and insulin secretion stimulated by glucose in mice with *SIRT6* overexpression. Their study indicated that *SIRT6* overexpression increases triglyceride clearance in the blood and reduces triglyceride production in adipose tissue. Mice with *SIRT6* overexpression subjected to a high-fat diet accumulated significantly less visceral fat, LDL cholesterol and triglycerides compared to wild-type mice. This suggests a protective role of SIRT6 against metabolic consequences of obesity caused by an improper diet [40]. Tang et al. [46] came to similar conclusions—*SIRT6* overexpression in the arcuate nucleus of the hypothalamus in mice reduced their body weight induced by a high-fat diet. A decrease in the weight of eWAT (epididymal white adipose tissue) and iWAT (inguinal white adipose tissue) and adipocyte size was also observed. Furthermore, the impairment of leptin activity in the POMC neurons of mice subjected to *SIRT6* neuron-specific knockout was demonstrated. These mice exhibited a predisposition to obesity and increased food consumption.

Increased SIRT6 levels were observed in in vivo models—in mice after fasting, rats after caloric restriction and in vitro—in cell cultures in a medium deprived of nutrients. The authors indicated that the increase in SIRT6 levels is caused by the stabilization of the SIRT6 protein, not an increase in *SIRT6* transcription. Moreover, p53 positively regulates SIRT6 protein levels under standard growth conditions but does not play a role in regulating *SIRT6* under caloric restriction [47]. It is widely known that calorie restriction diets (CR) slow down aging processes and may contribute to extending life [48]. In light of the above research, it can be suggested that the beneficial effects of a calorie-restricted diet are strongly related to increased SIRT6 stability, and the increased expression of the protein generated by factors other than starvation may exhibit similar effects to CR [47]. The role of SIRT6 in metabolism has been summarized in Table 1.

The “sirtfoods’’ diet, combing sirtuin-activating foods belonging to both Mediterranean and Asian diets, may be a promising dietary strategy in preventing chronic diseases, thereby ensuring healthy aging [49]. “Sirtfoods’’ can be found in: olive oil [50]; red wine [51]; grapes [52,53]; apple, strawberries, onion and cabbage [53]; soybeans [54]; tofu [55]; licorice [56]; shallot [56]. Pallauf et al. [49] suggested that omega-3 fatty acids, vitamins and antioxidants do not work in isolation. They should synergistically work to prevent chronic diseases.

## 5. Inflammation

The role of SIRT6′s involvement in inflammation is complex. This protein can have both pro- and anti-inflammatory properties, depending on the type of cells involved [36]. It has been suggested that SIRT6 acts as a pro-inflammatory agent by functioning as a lysine deacetylase, removing fatty acyl modifications from K19 and K20 of TNF-α and thereby promoting the secretion of TNF-α from the cell [57].

A characteristic feature of inflammatory diseases is endothelial dysfunction. Factors associated with this phenomenon include the increased release of pro-inflammatory cytokines, adhesion molecules and tissue matrix-degrading enzymes (collagenases and others). At the transcriptional level, this is regulated by the histone deacetylase SIRT6 through its action on the pro-inflammatory transcription factor nuclear factor-κB (*NF-κB*) [58]. In a study by Lappas et al. [59], the role of SIRT6 in regulating inflammation in endothelial cells was determined. They evaluated markers of inflammation in human umbilical vein endothelial cells (HUVECs) in the presence of lipopolysaccharide (LPS) as a model agent for initiating inflammation. LPS decreased *SIRT6* expression in HUVEC cells, which agrees with previous observations by other authors [60,61]. In contrast, *SIRT6* knockdown increased the expression of a number of pro-inflammatory cytokines (IL-1β, IL-6, IL-8), the COX-prostaglandin system, extracellular matrix remodeling enzymes (MMP-2, MMP-9 and PAI-1), the adhesion molecule ICAM-1 and angiogenesis-related growth factors (VEGF and FGF-2). Knockdown of *SIRT6* increased the expression of *NF-κB*. In contrast, *SIRT6* overexpression was associated with decreased *NF-κB* transcriptional activity. Thus, they indicated that the loss of *SIRT6* in endothelial cells is associated with the up-regulation of genes involved in the progression of inflammation and associated vascular remodeling. Hence, the authors conclude that the up-regulation of *SIRT6* expression is a potential pharmacological target for drugs that reduce the inflammatory destruction of blood vessels. Such action will have the potential to nullify the negative manifestations of cardiovascular disease, diabetes and a range of neurodegenerative diseases. An extensive discussion of this topic was carried out in the work of Guo et al. [62].

Earlier, the role of LPS was identified as a factor that decreases the expression of *SIRT6*. It is suggested that this mechanism is based on the weakening of NF-κB signaling through H3K39 deacetylation on chromatin. NF-κB is a key factor involved in regulating cell apoptosis, aging, inflammation and immune system function [63]. Hyperactive NF-κB signaling can therefore contribute to both normal and premature aging. Hence, the role of SIRT6 in reducing the activity of this factor is significant [64].

Balestrieri et al. [37] demonstrated reduced *SIRT6* expression and lower interstitial collagen content in cells obtained from homogenates of atherosclerotic plaques from the carotid arteries of people with diabetes compared to plaques obtained from people without diabetes. In addition, plaques obtained from patients with diabetes were characterized by greater inflammation and oxidative stress. Subsequent trials showed that atherosclerotic plaques obtained from diabetic patients treated with GLP-1 (glucagon-like peptide-1 receptor) drugs for 26 ± 8 months indicated higher expression of *SIRT6* and collagen, as well as a lower intensity of inflammation and oxidative stress than cells obtained from untreated patients. This indicates the involvement of SIRT6 in the inflammatory changes of atherosclerotic lesions in diabetes and the role of GLP-1 in modulating it. These results were further supported by observations in an in vitro model. Endothelial progenitor cells (EPCs) and endothelial cells (ECs) were used as model cells. Both types of cells treated with high glucose medium (25 mmol/L) in the presence of GLP-1 (100 nmol/liraglutide) showed higher expression of *SIRT6* and lower expression of nuclear factor-κB compared to control cells (only treated with high glucose). The results obtained by Balestrieri et al. [37] confirmed the controlling involvement of SIRT6 in the inflammatory pathways of diabetic atherosclerotic lesions. From the perspective of skin wellbeing, this is an important observation. The incidence of diabetic complications involving the skin and its appendages is high, and identifying treatment options to nullify the underlying pathomechanism, i.e., vascular destruction, is an important topic here.

The promising role of *SIRT6* regulation is indicated to support the treatment of: atherosclerosis [65]; diabetic atherosclerosis [66]; diabetes-associated inflammatory diseases [67]; diabetic nephropathy [68]; adipose tissue inflammation [69]; oxidative stress in the heart triggered by high-fat diet-induced obesity [70] and osteoarthritis [61,71].

## 6. Role in the Wound Healing Process

With age, the loss of homeostasis leads to a series of clinical consequences, including impaired wound healing. Macrophages play a key role in wound healing, as they are responsible for suppressing excessive inflammatory states, removing cell debris and initiating and coordinating tissue remodeling and regeneration [72,73]. Activated macrophages can be divided into two phenotypes: M1 (mainly involved in pro-inflammatory responses) and M2 (responsible for anti-inflammatory actions) [19,73].

SIRT6 is considered a key factor in switching macrophage phenotypes and migratory responses, and therefore, it is believed to play a role in the wound healing process [18]. To illustrate the effect of SIRT6 on wound healing, Koo et al. [18] created a mouse model with *SIRT6* deletion (mS6KO). Wounds were excised on the dorsal skin of female mS6KO and control group mice (WT). Slower wound healing was observed in mS6KO mice. Reduced collagen content in the granulation tissue of mS6KO mice was demonstrated, as well as suppressed angiogenesis gene activity in mS6KO mice. Moreover, increased M1 macrophage infiltration with decreased M2 macrophage numbers, as well as more pro-inflammatory cytokines TNF-α, IL-1β and IL-6 were noted. This suggests that the specific deficiency of SIRT6 in bone marrow cells has an impact on the temporal response of the wound healing process by inhibiting epithelial regeneration, angiogenesis and collagen deposition. Thandavarayan et al. [22] arrived at similar conclusions. In a mouse model with diabetes, they showed that SIRT6 deficiency reduced *VEGF* expression, increased the expression of pro-inflammatory markers (including TNF-α, IL-1β) and intensified oxidative stress when siRNA against *SIRT6* was administered. Yang et al. [74] described the molecular mechanism that connects SIRT6 and angiogenesis.

Furthermore, exogenous substances that are associated with the modification of the intensity of sirtuin expression can affect changes in the rate of wound healing. This issue has been the subject of several research projects [75,76,77,78,79].

Resveratrol, a sirtuin activator, by increasing the proliferation of keratinocytes, accelerates wound healing, while sirtinol, a sirtuin inhibitor, will delay their closure. This suggests the role of administering natural or synthetic *SIRT6* activators as factors accelerating wound healing processes [77].

Flavonoids are polyphenolic secondary metabolites synthesized by fungi and plants*;* possessing various pharmacological activities. It has been shown that compounds from this group modulate the activity of SIRT6 by altering its structure.

SIRT6 activators presumably bind near the acetylated peptide substrate binding site. Inhibitors, on the other hand, are likely to bind in a way that disrupts NAD^+^ binding. Cyanidin is indicated as the strongest SIRT6 activator [80]. Other activators include: N-acetylamines [81], icariin [82], ergothioneine [83]. Significant inhibitory power for SIRT6 activity has been shown for galloylated catechins [80]. Compounds that will exhibit inhibitory activity on SIRT6 activity include quercetin, vitexin [84], peptides containing netioacetylated lysine [85] and EX-527 [86]. The role of SIRT6 in the wound healing process is shown in Figure 3.

## 7. Role of Physical Activity

Appropriate levels of physical activity are an important element in preventing premature disability, the progression of various lifestyle diseases and aging. Increasing the endurance capacity of skeletal muscles is a new strategy in the fight against metabolic diseases associated with obesity, and this can be achieved by shifting skeletal muscle fibers towards slow-twitch, oxidative fibers. Aging leads to a gradual decrease in physical activity and disturbances in energy homeostasis. Numerous studies have focused on the correlation between changes in activity levels and *SIRT6* expression [87,88,89,90,91,92] (Figure 4).

The expression of *SIRT6* is increased in chronically exercising individuals. Introducing resistance training as a modification of current physical activity allows for an increase in SIRT6 concentration in the blood of older men [65]. A factor that increases *SIRT6* expression is significant weight loss achieved through diet therapy and exercise in morbidly obese individuals [88].

SIRT6 regulates aging and a series of metabolic processes. Roichman et al. [89] showed that *SIRT6* overexpression leads to a reduction in frailty and an extension of lifespan in both male and female mice. Older transgenic *SIRT6* mice maintained normal glucose production in the liver and glucose homeostasis by improving the utilization of two major gluconeogenic precursors, lactate and glycerol. Therefore, *SIRT*6 overexpression protected older individuals from impaired carbohydrate homeostasis. The authors indicated that the mechanism of this beneficial phenomenon is associated with increased gene expression for proteins involved in gluconeogenesis in the liver, de novo synthesis of NAD^+^ (which is also the basis for the beneficial effects of physical training) and systemically increases the release of glycerol from adipose tissue. SIRT6 optimizes energy homeostasis in older age, which is associated with the carbohydrate theory of aging. It was also noted that it is precisely SIRT6 that mediates the beneficial effect of regular physical activity in preventing insulin resistance [90].

The study by Song et al. [91] indicated that SIRT6 is crucial in regulating the configuration of muscle fibers toward the oxidative type. They also indicated that a SIRT6 activator (MDL801) could produce the same effects as physical training. Genetic inactivation of SIRT6 in skeletal muscle decreased, while its transgenic overexpression increased mitochondrial oxidative capacity and exercise performance in the model animals studied. The authors indicated that the ablation of *SIRT6* in skeletal muscle led to reduced exercise performance. The authors tested the hypothesis that *SIRT6* induction may be mediated by endocrine/paracrine signaling (Il-6 enhanced S*IRT6* expression) and/or a stimulus effect (stimulating the muscle with electrodes also enhanced *SIRT6* expression). *SIRT6* ablation also reduces the number of mitochondria; knockout muscles also had significantly more mitochondria with abnormal structures. Reduced mitochondrial DNA content and suppressed expression of genes related to mitochondrial biogenesis were observed. The consequence was the suppression of basal, ATP-related and maximal respiration in *SIRT6* knockout mice observed in this study. *SIRT6* ablation impairs the ability to meet the cell’s energy needs through oxidative phosphorylation [91].

The mechanism linked *SIRT6* to the downregulation of *Sox6*, a key repressor of the free-fiber-specific gene, by increasing *CREB* transcription. This mechanism demonstrates how suppressed *SIRT6* expression links to the mitochondrial theory of aging in how the beneficial effects of regular physical activity on life extension and quality of life binding can be explained.

It has also been indicated, in studies conducted on young men, that even a single bout of aerobic exercise (at 80% of peak oxygen uptake) will affect the expression of *SIRT6* and telomerase, the enzyme responsible for maintaining the proper length of telomeres protecting the ends of DNA strands [92]. This observation will again allow for linking SIRT6 to the telomeric theory of aging. 

## 8. Conclusions

Research suggests that the activation of *SIRT6* and its role in various biochemical processes may have a positive impact on health and longevity. Several pathomechanisms associated with accelerated aging of the body as a whole, as well as individual systems, could be inhibited by an increased expression of this protein. Modulating SIRT6 activity can therefore be a valuable tool for supporting the treatment of age-related diseases and improving quality of life, thereby contributing to longevity. In addition to well-known lifestyle modifications, such as caloric restriction and intermittent fasting, research is being conducted to identify new pharmacologically active molecules and dietary components with recognized roles in promoting longevity. Furthermore, physical activity might be a valuable tool for modifying and/or modulating SIRT6 for health and longevity. As seen in relation to the presented results in this review, not all aspects related to the activity of SIRT6 are fully understood. Hence, there is a necessity and need for systematic, controlled research into this continually emerging area of investigation.

## Figures and Tables

**Figure 1 ijms-24-09655-f001:**
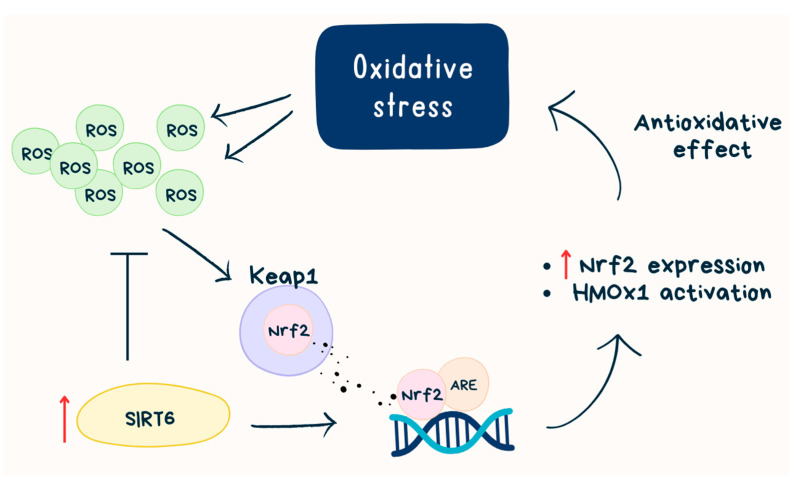
SIRT6 and Nrf2 path. ROS—reactive oxygen species; SIRT6—sirtuin 6, Keap1—Kelch-like ECH-associated protein 1, Nrf2—erythroid 2-related factor 2; ARE—antioxidant response elements; HMOX1—heme oxygenase 1.

**Figure 2 ijms-24-09655-f002:**
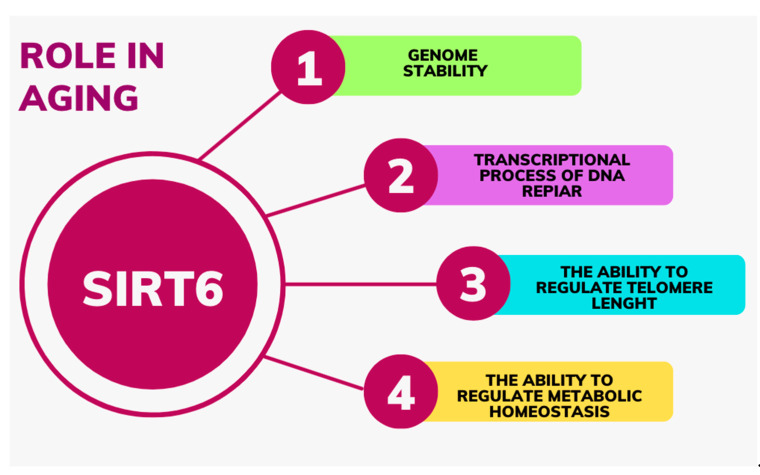
The role of the SIRT6 in aging.

**Figure 3 ijms-24-09655-f003:**
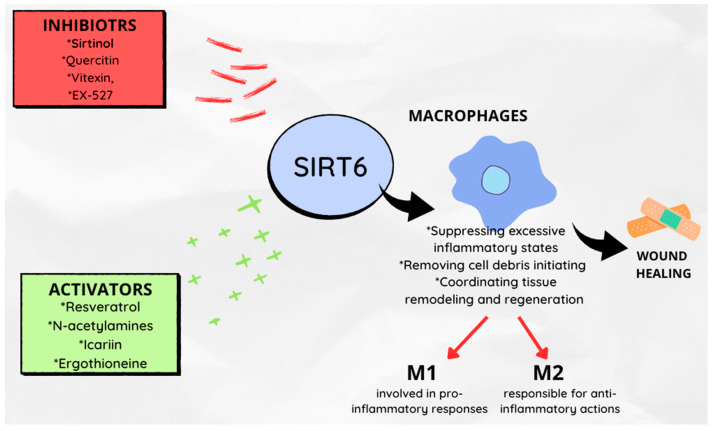
The role of SIRT6 in the wound healing process.

**Figure 4 ijms-24-09655-f004:**
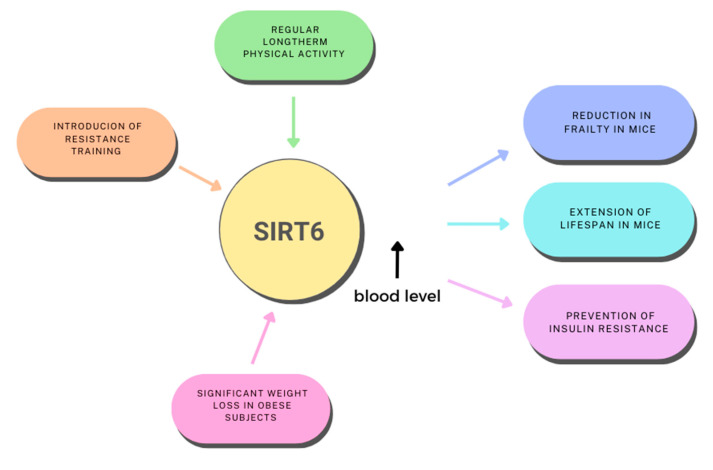
Modifying role of physical activity and the influence of SIRT6 on metabolism based on: [85,88,89,90].

**Table 1 ijms-24-09655-t001:** Role of the SIRT6 in metabolic activity regulation.

SIRT6 and Metabolic Activity	
Obesity	SIRT6 plays a protective role against the metabolic consequences of diet-induced obesity, which suggests a potentially beneficial effect of SIRT6 activation on age-related metabolic diseases [40]. In obese patients, the expression of SIRT6 is reduced. It suggests that SIRT6 is an attractive therapeutic target for treating obesity and obesity-related metabolic disorders [45].
Fat metabolism	SIRT6 plays a critical role in fat metabolism, and may therefore be a potential target in the treatment of liver diseases characterized by lipid accumulation [42].
Carbohydrate metabolism	Activation of hepatic by SIRT6 may be therapeutically useful for treating insulin-resistant diabetes [11]. SIRT6 may be useful to improve insulin secretion in diabetes [44].
Energy balance	SIRT6 is an important molecular regulator for POMC neurons to promote negative energy balance [46].
Other	SIRT6 appears to function as a corepressor of the Hif1α—a critical regulator of nutrient stress responses [41]. Expression of SIRT6 increased upon nutrient deprivation in cultured cells. The increase in SIRT6 levels is due to the stabilization of the SIRT6 protein, and not via an increase in SIRT6 transcription [47].

## Data Availability

Not applicable.

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
