# Peer review of "The Role of Increased Expression of Sirtuin 6 in the Prevention of Premature Aging Pathomechanisms"

_ijms, 2023, doi:10.3390/ijms24119655_

Round 1
Reviewer 1 Report
Dear authors
The manuscript named “The role of increased expression of Sirtuin 6 in the prevention 2 of premature aging pathomechanisms” aimed to show the association Sirtuin 6 with aging associated pathological mechanism by literature review. The authors related the involvement of Sirt 6 on several molecular mechanism that contributed to health aging. The study is very interesting, and I believe that it deserves be published, with a few corrections detailed bellow:
In introduction section, there no justified why the authors selected Sir 6 to review instead the other ones.
In the review, the authors could add some studies about Sir 6 and nutritional habits, as way to present the food categories associated with increase levels of Sirtuin in organisms. In addition, the title of manuscript suggested that pathological mechanisms will be included in this review, however they are superficial mentioned. Current studies about atherosclerosis, osteoarthritis, obesity, and diabetes must be included.
At end, authors could use more updated references to address the impact of Sirt6 on aging.
Regards
Author Response
Our answers can be found in the appendix

Reviewer 2 Report
Reviewer Recommendation and Comments for Manuscript Number:
ijms-2381525
“The role of increased expression of Sirtuin 6 in the prevention 2 of premature aging pathomechanisms”
These are my general/specific comments:
GENERAL:
The current manuscript is a very interesting and well-synthesized review focusing on the role SIRT6 has in and on aging. The manuscript was a delight to review; and, indeed, the author’s comprehension and discussion of both the historical and contemporary research with respect to sirtuins, specifically SIRT6, is thorough. The importance of and need for systematic, controlled research into this continually emerging area of investigation is eluded to, yet this is more of a passing point that should be emphasized in the Conclusion. Overall, the manuscript has relatively minor editorial (grammar, language, etc.,) modifications that should be addressed easily. There are additional limitations to the manuscript in its current form, which are commented on and should be modified by the authors. Also, Figure 1 text and resolution should be improved (i.e. enlarged for readability). Even though there are limitations in the manuscript’s present form, the authors should be able to address these points for full consideration of acceptance.
SPECIFIC:
TITLE
- None
ABSTRACT
- Front page (Page 1), Lines 17-19: Move sentence “This study discusses…” to the end of the Abstract
INTRODUCTION
- Overall and throughout the manuscript the authors should be consistent with the specific verbiage used to describe sirtuins; please utilize SIRT6 throughout after the introduction of the acronym. If the authors do utilize multiple acronyms (i.e. SIRT6, Sirt6 for differentiating between protein and/or gene, then this should be explicitly stated)
- Page 2, Lines 47-49: Delete sentence “The conserved enzyme…” This is redundant
- Page 2, Line 69: Change “on” to “in”
Structure
- Page 2, Line 81: Bold “Activation and reactivation of gene expression”
- Page 2, Line 86: Change “IRT6” to “SIRT6”
Aging
- Page 3, Line 101: Change “irreversible” to “inevitable”
- Page 3, Line 106: Change “believed” to “hypothesized”
- Page 3, Line 135: Add “this” prior to “Betry”
- Page 4, Line 172: Please clarify that the discussion in this paragraph all relates and references to Sharma et al. [6]; if not, a Reference is needed following the last sentence of this paragraph
- Page 4, Line 180: Debold “An”
Metabolic Activity
- Page 6, Line 227: Add “white” prior to “adipose”
- Page 6, Line 239: Change “overexpression” to “increased expression”
Inflammation
- Page 6, Line 243: Change “…SIRT6 in controlling inflammation…” to “…SIRT6’s involvement in inflammation…”
- Page 6, Line 253: Delete “which is also a histone deacetylase,”
Role in the wound healing process
- Page 8, Line 314: Change “slowdown” to “temporal response”
Role of physical activity
- Page 8, Line 344: Add “comma” following “slow-twitch”
- Page 8, Line 349-351: Please rephrase/rewrite this sentence for clarity, for as written it is very confusing
- Page 9, Line 369: Please clarify that the discussion in this paragraph all relates and references to Song et al. [69]; if not, a Reference is needed following the last sentence of this paragraph
Conclusions
- The paragraph does not include that physical activity/exercise may present as a valuable tool for modifying and/or modulating SIRT6 for health and longevity. This should be included for emphasis here; especially, given that it is commented upon in the Abstract as well as the page/paragraph preceding the Conclusions
REFERECES
- Please refer to comment(s) above concerning manuscript text/references
FIGURE LEGENDS/FIGURES
- Page 7, Line 303-304: Figure 2 text needs to be enlarged and the resolution should be enhanced for clarity. It is difficult to read and blurry
Quality of English is adequate.
Author Response

(The authors gave the same response as above.)

Reviewer 3 Report
1. Minor issue on page 1: prokaryotes need not be capitalized.
2. In the first section, expanding discussion on the discovery of SIRT proteins would be helpful.
3. On page 2, SIRT6 is misspelled.
4. Table 1 could be covered in the text. The table does not add significantly to the manuscript.
5. There are two figures labelled as Figure 2. The first does not add significantly to the manuscript. Overall, the figures should be redone to better summarize the literature on the topic.
6. Elaborating on signaling pathways, including those with Nrf2, would strengthen the manuscript. Provide additional details and integrate these concepts within each section.
Minor editing needed.
Author Response

(The authors gave the same response as above.)

Round 2
Reviewer 2 Report
While the authors should be commended on their revised manuscript that now may be considered fully for acceptance for publication, there are two minor typographical/editorial comments that this review makes: (1) A "period" is needed for the final sentence of the Abstract, and (2) on Page 7 the initial quotation mark should be modified (it is currently a subscript) on "sirtfoods".
Reviewer 3 Report
All issues have been addressed.